# Seven-month kinetics of SARS-CoV-2 antibodies and role of pre-existing antibodies to human coronaviruses

Natalia Ortega [1,16], Marta Ribes [1,16], Marta Vidal[1], Rocío Rubio [1], Ruth Aguilar [1], Sarah Williams[1], Diana Barrios[1], Selena Alonso [1], Pablo Hernández-Luis [2,3], Robert A. Mitchell [1], Chenjerai Jairoce[1,4], Angeline Cruz [1], Alfons Jimenez [1,5], Rebeca Santano [1], Susana Méndez [1], Montserrat Lamoglia [1,6], Neus Rosell[1], Anna Llupià [1,7], Laura Puyol[1], Jordi Chi [1], Natalia Rodrigo Melero[8], Daniel Parras[2], Pau Serra[2], Edwards Pradenas [9], Benjamin Trinité[9], Julià Blanco [9,10], Alfredo Mayor [1,4,5], Sonia Barroso[11], Pilar Varela[11], Anna Vilella[1,5], Antoni Trilla [1,5,12], Pere Santamaria[7,13,14], Carlo Carolis [8], Marta Tortajada[11], Luis Izquierdo [1], Ana Angulo[2,3], Pablo Engel[2,3], Alberto L. García-Basteiro [1,4,15], Gemma Moncunill [1,17✉] & Carlota Dobaño [1,5,17✉]

Unraveling the long-term kinetics of antibodies to SARS-CoV-2 and the individual characteristics influencing it, including the impact of pre-existing antibodies to human coronaviruses causing common cold (HCoVs), is essential to understand protective immunity to COVID-19 and devise effective surveillance strategies. IgM, IgA and IgG levels against six SARS-CoV-2 antigens and the nucleocapsid antigen of the four HCoV (229E, NL63, OC43 and HKU1) were quantified by Luminex, and antibody neutralization capacity was assessed by flow cytometry, in a cohort of health care workers followed up to 7 months ($N = 578$). Seroprevalence increases over time from 13.5% (month 0) and 15.6% (month 1) to 16.4% (month 6). Levels of antibodies, including those with neutralizing capacity, are stable over time, except IgG to nucleocapsid antigen and IgM levels that wane. After the peak response, anti-spike antibody levels increase from ~150 days post-symptom onset in all individuals (73% for IgG), in the absence of any evidence of re-exposure. IgG and IgA to HCoV are significantly higher in asymptomatic than symptomatic seropositive individuals. Thus, pre-existing cross-reactive HCoVs antibodies could have a protective effect against SARS-CoV-2 infection and COVID-19 disease.

---

A full list of author affiliations appears at the end of the paper.

Coronavirus disease 2019 (COVID-19), caused by the Severe Acute Respiratory Syndrome Coronavirus 2 (SARS-CoV-2), has already caused a loss of 3.2 million lives globally (23 June)[1]. Since its emergence, a key priority has been the understanding of the kinetics and protective role of the immune response in the population, to assess the degree of exposure in serosurveys and to understand immunity to the virus. This knowledge guides vaccine development, selection of donors for hyperimmune serum-transfusion therapies, and combining antigens with the highest immunogenic and neutralizing capacity to improve surveillance interventions.

Longitudinal studies assessing SARS-CoV-2 antibody kinetics have found that IgA and IgM peak between week 3 and 4 post symptoms onset (PSO) and wane thereafter, with IgA persisting longer than IgM[2–7]. IgA and IgM seroreversion were estimated between days 71 and 49, respectively[8], but IgA has also been found to remain detectable 6 months post infection and to be less affected by the decay than IgM[9,10]. Several studies have observed relatively stable levels of IgG to the spike (S) protein after three[6,11], four[12,13], and six to eight months[2,9,14–16]. However, others reported that IgG only lasted around 3-4 months PSO[17,18]. Many studies consistently observe that IgG to the nucleocapsid (N) protein, found inside the virus or infected cells, decay faster than IgG to S, being a marker of a more recent infection but less sensitive for assessing population seroprevalence[2,13–20]. While antibodies targeting N protein are unlikely to directly neutralize SARS-CoV-2, those targeting S, responsible for the interaction with the ACE2 receptor in the host cells, are considered the main neutralizers[21]. Studies up-to-date point that neutralizing antibodies (nAbs) strongly correlate with antibody titers to S[16,19–22] and also positively correlate with increased disease severity[23–26].

Understanding the extent of antibody cross-reactivity with other human coronaviruses (HCoV) is important to elucidate the impact of such pre-existing antibodies on COVID-19 immunity. Four low-pathogenic HCoV causing common cold have circulated among humans for at least 100 years: the alphacoronaviruses 229E and NL63, and the betacoronaviruses OC43 and HKU1. They account for about 10% of all acute respiratory tract infections, and thus, a substantial proportion of the global population is expected to carry antibodies against them[27,28], although their protective immunity might be short-lasting[29]. Previous studies found some cell-mediated[30,31] and antibody cross-reactivity of HCoV immune responses with SARS-CoV-2[32–34]. Regions within N and S antigens with high amino acid homology between SARS-CoV-2 and HCoV are potential targets of cross-reactive antibodies[33–36], and could exert cross-protective effects against SARS-CoV-2 infection and/or disease. Prior studies have not found protection against infection, as participants with recent documented infection with an endemic HCoV had similar rates of SARS-CoV-2 acquisition than those without recent HCoV infection[37–39]. Regarding anti-disease protection, COVID-19 patients with a recent HCoV diagnosis had statistically significant lower odds for COVID-19 intensive care unit admission and death[39], but other studies did not find any association between confirmed prior history of seasonal HCoVs and COVID-19 severity[37,38]. Some recent studies have suggested that this pre-existing immunity would not confer cross-protection but, rather, be responsible for an immunological imprinting or 'original antigenic sin', a phenomenon well studied for influenza virus infections. This suggests that the immune system privileges recall of existing memory responses -in this case of HCoV-, in detriment of stimulating de novo responses -here to SARS-CoV-2- leading to poor outcomes or severe disease[31,32]. The possibility of antibodies to HCoVs acting as antibody-derived enhancement (ADE) has also been reviewed and the most recent evidence shows no clinical, in vitro or animal evidence[40,41]. Disentangling

the role of pre-existing HCoVs antibodies on anti-SARS-CoV-2 responses may have implications in the deployment of potentially effective vaccines, as well as for the interpretation of serological studies.

At the beginning of the pandemic, healthcare workers (HCW) were considered to be at a higher risk of SARS-CoV-2 infection than the general population, although there is now evidence that seroprevalence is similar when using adequate personal protective equipment. We previously observed 9.3% (95% CI, 7.1–12.0) SARS-CoV-2 seroprevalence in a cohort of 578 HCW from Hospital Clínic in Barcelona (HCB) between March-April 2020[42], and of 14.9% after a month follow-up[4], based on the detection of antibodies to one antigen (receptor binding domain, RBD). IgA, IgM, and IgG levels declined after 3 months with antibody decay rates of 0.12, 0.15, and 0.66 respectively[4].

In the present study, we aimed to characterize the antibody kinetics and neutralization capacity between March and October 2020 at four cross-sectional surveys and estimate the seroprevalence in the same cohort of HCW. For this analysis, we measured IgM, IgG, and IgA isotypes against an expanded panel of six SARS-CoV-2 antigens and tested cross-reactivity with the N antigen of the four endemic HCoVs (HKU1, 229E, OC43 and NL63) to assess its potential impact on COVID-19 protection.

## Results

**Seroprevalence, seroconversions, and seroreversions.** From the initial cohort, 507 individuals participated in a fourth visit (M6) six months after baseline (12.3% lost to follow-up). The mean age was 42.7 years (SD: 11.2) and 72% were female. Full demographic characteristics at baseline (M0), one (M1), and three (M3) month follow-up visits were as described[4,42] (Supplementary Table 1).

Samples collected at M0, M1, and M3 were re-tested with a wider panel of antigens along with M6 samples. The seroprevalence for either IgM and/or IgG and/or IgA was 13.5% at M0, 15.6% at M1, and 16.4% at M6 (Supplementary Table 2). Newly detected SARS-CoV-2 infections increased by 22, 9 by rRT-PCR and 13 by serology, at M6 compared to visit M1. When considering rRT-PCR and serology data, 84 out of 578 participants (14.5%, 95% CI 11.8–17.7%) had evidence of infection at M0 by serology or rRT-PCR, 91/566 (16.1%, 95% CI 13.1–19.4%) at M1 and 91/507 (17.9%, 95% CI 14.7–21.6%) at M6. The cumulative prevalence of infection was 16.8% (95% CI 13.8–20.1%) and 19.6% (95% CI 16.4–23.0%) at M1 and M6, respectively. Unlike seropositive proportions, we had a relatively stable number of undetermined results over time, 48 (8.3%), 52 (9.1%), and 37 (7.3%) participants at baseline, M1 and M6, respectively. Sixty-seven out of the 119 participants (56.3%) with any evidence of infection had a positive rRT-PCR (Supplementary Table 2).

At visits M1 and M3, we mainly observed seroreversions of seropositive individuals at M0 for IgA and IgM to all antigens (30% and 24.5%, respectively). Hardly any participant seroreverted from M3 to M6 (Supplementary Table 2). Overall, there were 9 participants who were seronegative and previously had a positive rRT-PCR, 32 to 197 days prior to sample draw. Three of these HCWs were asymptomatic.

Physicians and psychologists had 50% lower odds of infection (OR 0.49, 95% CI 0.27–0.85) than nurses and other auxiliary health professionals (Supplementary Table 3). Age, sex, and other variables were not found to be associated with SARS-CoV-2 infection. Sixty-nine percent of the infections were symptomatic and a single participant required hospitalization in our cohort.

**Kinetics of SARS-CoV-2 antibodies up to 7.7 months PSO.** Levels of SARS-CoV-2 antigen-specific isotypes (IgM, IgA, IgG) were plotted against time with up to four observations

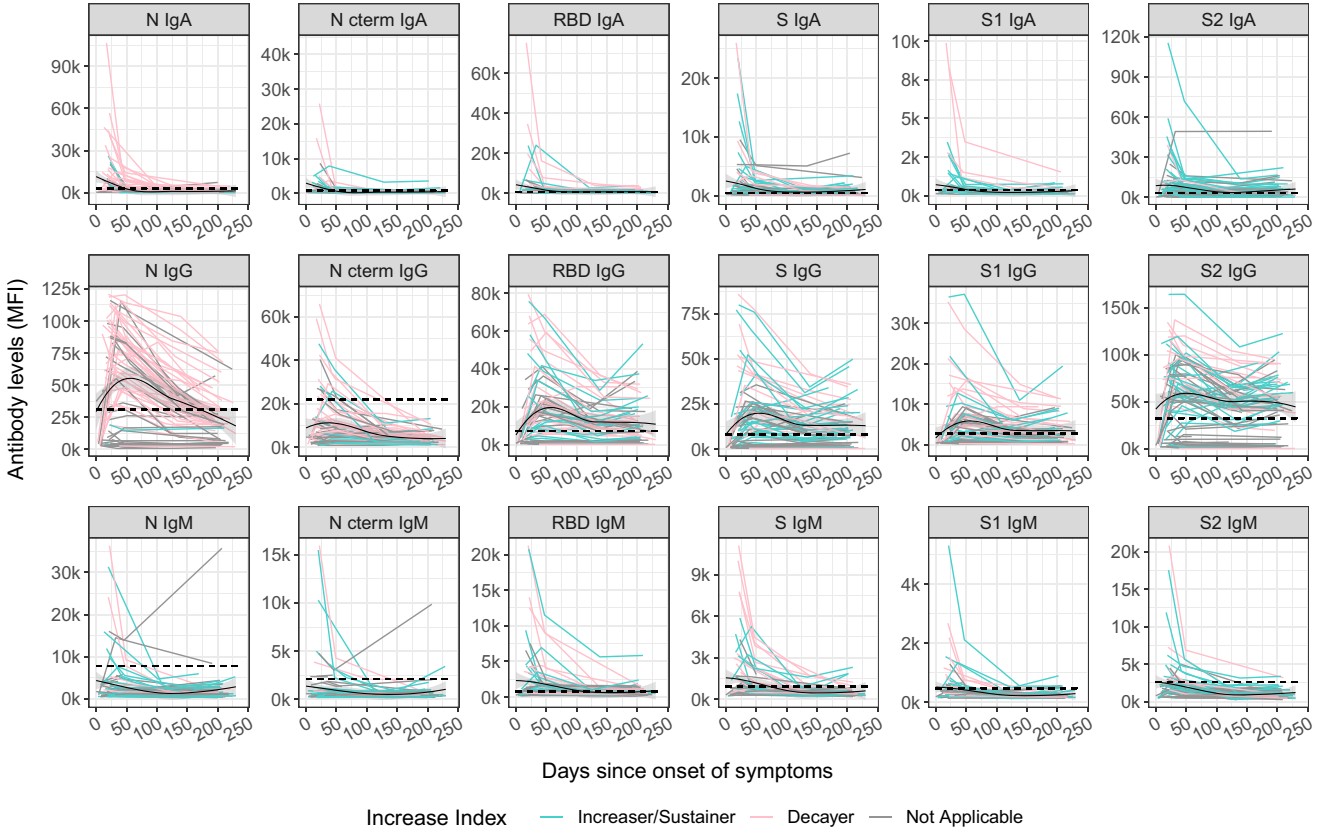

**Fig. 1 Kinetics of SARS-CoV-2 antibody levels since onset of symptoms.** Levels (median fluorescence intensity, MFI) of IgA, IgG, and IgM against each antigen (Nucleocapsid full-length protein (N), and its C-terminal domain, the Receptor Binding Domain (RBD), full S protein and its subregions S1 and S2) measured in 235 samples from 76 symptomatic participants collected in up to four time points per participant (paired samples joined by lines). The black solid line represents the fitted curve calculated using the LOESS (locally estimated scatterplot smoothing) method. Shaded areas represent 95% confidence intervals. Dashed line represents the positivity threshold. Participants were grouped based on their antibody levels at M6 compared to the previous visit, individuals were labeled for each isotype-antigen pair as "Decayers" (pink) when the ratio of antibody levels between both visits was <1 and as "Sustainers/Increasers" (light blue) when the ratio was ≥1 and gray when the classification was not applicable.

with a maximum 7.7 months PSO, in a total of 235 samples from 76 symptomatic participants (Fig. 1). Kinetic curves were very similar when plotted against days since positive rRT-PCR in participants who were asymptomatic or symptomatic (Supplementary Fig. 1).

IgA or IgM peaked within the first month PSO, while IgG peaked around day 50 PSO. SARS-CoV-2 IgG levels were generally steady for S antigens (S, S1, S2, and RBD) and for IgA up to 230 days PSO (71% and 69% of the participants remained seropositive six months PSO, respectively), and waned at a clearly slower rate than IgM (34% of the participants remained seropositive) and IgG to N-related antigens (26% of the participants remained seropositive).

Antibody levels were observed to increase from ~150 days PSO onwards (Fig. 1). To further explore this, we grouped participants based on their antibody levels at M6 compared to the previous visit (M1 or M3). We only considered participants who had already shown a decrease in antibodies after the peak response. We, therefore, calculated an "antibody increase index" between both visits for each antigen-isotype combination and labeled the individuals as "decayers" when the ratio of antibody levels between both visits was <1, and as "sustainers/increasers" when the ratio was ≥1, in line with the methodology by Chen et al.[17]. Increased levels were observed in all antigen-isotype combinations (Fig. 1). Most sustainers/increasers had a boost for more than one antigen-isotype pair, as assessed by a Venn diagram

(Supplementary Fig. 2). Levels at seroconversion visit were higher in decayers than sustainers/increasers, being statistically significant for N IgG, S2 IgG, and S1 IgM (Supplementary Fig. 3a). There was no association of the antibody increase index or being a sustainer/increaser or a decayer with age. We observed a trend towards having a higher antibody increase index, mainly for IgG, in participants who reported current or past symptoms at M6 since the last visit, several months after COVID-19 disease recovery (Supplementary Fig. 3b). We also identified a trend towards a higher antibody increase index in participants with a shorter duration of symptoms (<10 days) compared to those who had symptoms for >10 days (Supplementary Fig. 3c).

**Kinetics of neutralizing antibodies.** Plasma neutralizing capacity measured as RBD-ACE2-binding inhibition generally increased between the day of onset of symptoms until day 80 and remained stable thereafter up to 250 days PSO (Fig. 2). We correlated the antibody neutralizing capacity and levels at the different study visits. At the first cross-sectional visit (M0, mean days PSO = 20) levels of all three Ig isotypes against RBD and S antigens positively correlated with neutralization capacity ($r_s = 0.19–0.32$, $p < 0.05$), while the correlation between antibody levels against N and RBD-ACE2 neutralization did not reach statistical significance (Fig. 3A). At the fourth cross-sectional visit (M6, mean days PSO = 200), IgM levels to any antigen did not correlate with neutralization percentage, whilst IgG and IgA levels against all six

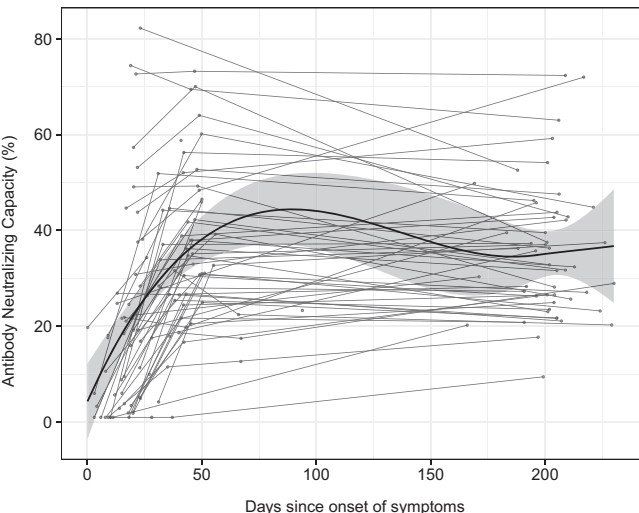

**Fig. 2 Longitudinal antibody neutralizing capacity.** Antibody neutralizing capacity, as a percentage of RBD-ACE2 binding inhibition in plasma samples from 64 symptomatic participants collected in three serial visits (M0, M1, and M6) represented as days after symptom onset. Paired samples are joined by gray lines. The black solid line represents the fitted curve calculated using the LOESS (locally estimated scatterplot smoothing) method. Shaded areas represent 95% confidence intervals.

antigens showed moderate to strong correlations ($r_s = 0.24–0.76$, $p < 0.05$), with higher correlations for S antigens (Fig. 3B). We performed a PCA for all antigen-isotype pairs (Supplementary Fig. 4) and the first five components, explaining 75.12% of the variance were included as predictors in a model with neutralizing capacity as an outcome ($p < 0.05$, adjusted $R^2$ 0.575). Component 1 and 5 were significantly associated with neutralizing capacity (Supplementary Table 4). In these components, S and S1 IgG, and S2 IgM, contributed to an increase in the neutralization activity, whilst N C-terminal IgG negatively influenced it ($p < 0.001$). We observed that antibodies to S antigens were highly contributing to the prediction of the neutralization percentage (component 1, longer vectors).

We did not find any significant difference in neutralizing capacity between sustainers/increasers and decayers. The neutralizing capacity was also not associated with the antibody increase index, except for IgM increase index that inversely correlated with the neutralization percentage at M0 and after six months PSO (Supplementary Fig. 5).

**Cross-reactivities of SARS-CoV-2 with endemic HCoV.** Pre-pandemic plasma samples had some antibody reactivity against SARS-CoV-2 antigens, particularly against N protein, and levels of antibodies against N from SARS-CoV-2 positively correlated with antibodies to HCoV N antigens (to a lesser extent for IgM), indicating cross-reactivity between them (Supplementary Fig. 6). The amino acid pairwise similarities and identities of full-length SARS-CoV-2 N protein and seasonal HCoVs are 36% and 26.4% to 229E, 39.1% and 27% to NL63, 48.1% and 35.7% to OC43 and 47% and 35.2% to HKU1[36]. 

Therefore, we analyzed the antibody levels against HCoV N antigens prior and after SARS-CoV-2 infection in the 33 participants who seroconverted during the study period. While some participants showed stable anti-HCoV N antibody levels, a general upward trend was observed. IgG to 229E significantly increased after SARS-CoV-2 seroconversion. Not all seroconverters had an increase in levels, supporting a back-boost of N HCoV beyond cross-reactivity (Supplementary Fig. 7).

We investigated whether having higher baseline anti-HCoV N antibody levels could be protective against SARS-CoV-2 infection. Overall, we observed a consistent trend towards higher baseline IgG levels to alpha-HCoV 229 ($p = 0.06$) and NL63 ($p = 0.15$) in participants who did not seroconvert compared to seroconverters, although these differences did not reach statistical significance (Fig. 4A). We assessed whether having higher anti-HCoV N antibody levels prior to infection could confer protection against COVID-19 symptoms in participants who seroconverted during the study period. Although statistical significance was only reached for IgA against OC43, we observed a common trend towards higher levels of anti-HCoV N IgA and IgG in asymptomatic than symptomatic SARS-CoV-2 seropositive participants (Fig. 4B). Consistently, levels of IgG against NL63 experienced a higher fold-increase after SARS-CoV-2 infection in asymptomatic than symptomatic seroconverters (Fig. 4C), suggesting that a back boost -beyond cross-reactivity-in anti-HCoV antibody levels could confer disease-protective immunity. In line with this finding, seropositive asymptomatic participants had significantly higher IgG levels against all four HCoVs than symptomatic participants in the first visit after SARS-CoV-2 positivity (Fig. 4D). In contrast, anti-SARS-CoV-2 N antibody levels were higher in symptomatic seropositive participants ($p < 0.05$) (Fig. 4E).

Finally, we tested whether baseline anti-HCoV antibody levels impacted de novo production of antibodies to SARS-CoV-2. To test this hypothesis, the increase in anti-N SARS-CoV-2 antibody levels from baseline to seroconversion for the three isotypes were correlated with the anti-N HCoVs antibody levels at baseline (adding up levels of isotypes). Overall, we observed a statistically significant inverse relationship between anti-HCoV IgG and IgA baseline levels and the increase of SARS-CoV-2 antibodies ($r_s = -0.35$, $p < 0.05$; $r_s = -0.18$, $p < 0.05$; respectively) (Supplementary Fig. 8). This suggests that pre-existing antibodies against the four HCoV N induced a lighter de novo production of antibodies against SARS-CoV-2 N.

## Discussion
We report a longitudinal study assessing the antibody response to a wide panel of antigens from SARS-CoV-2 and HCoV, up to 7.7 months after infection, and we show evidence of COVID-19 protection by pre-existing HCoV antibodies. This is important to track the evolution of the immunity in asymptomatic and mild/moderate cases, particularly in an indispensable population like HCW, and to understand why some people may be less affected by COVID-19. A strength of the present study is the availability of sequential sampling within a random cohort including asymptomatic and symptomatic subjects.

Importantly, we observed a trend towards higher levels of antibodies against HCoVs N proteins at baseline in those participants who did not become infected with SARS-CoV-2, suggesting some level of cross-protection against infection. Moreover, asymptomatic SARS-CoV-2 seropositive participants tended to have higher anti-HCoV N IgA and IgG levels prior to seroconversion than symptomatic participants, suggesting cross-protection against disease. In addition, asymptomatic seropositive participants had higher anti-HCoV N IgG levels after infection than symptomatics, pointing towards a disease-protective back-boost of anti-HCoV antibodies. Combined with the observation that higher baseline anti-HCoV N antibody levels correlated with less de novo anti-SARS-CoV-2 N antibody production, we propose a protective effect of previous exposure to HCoVs, which could be the result of a diminished exposure (decreased viral load) due to the suggested protective role of anti-HCoV antibodies. Other studies have reported a lack of anti-disease cross-protection[37–39]; and some studies have associated

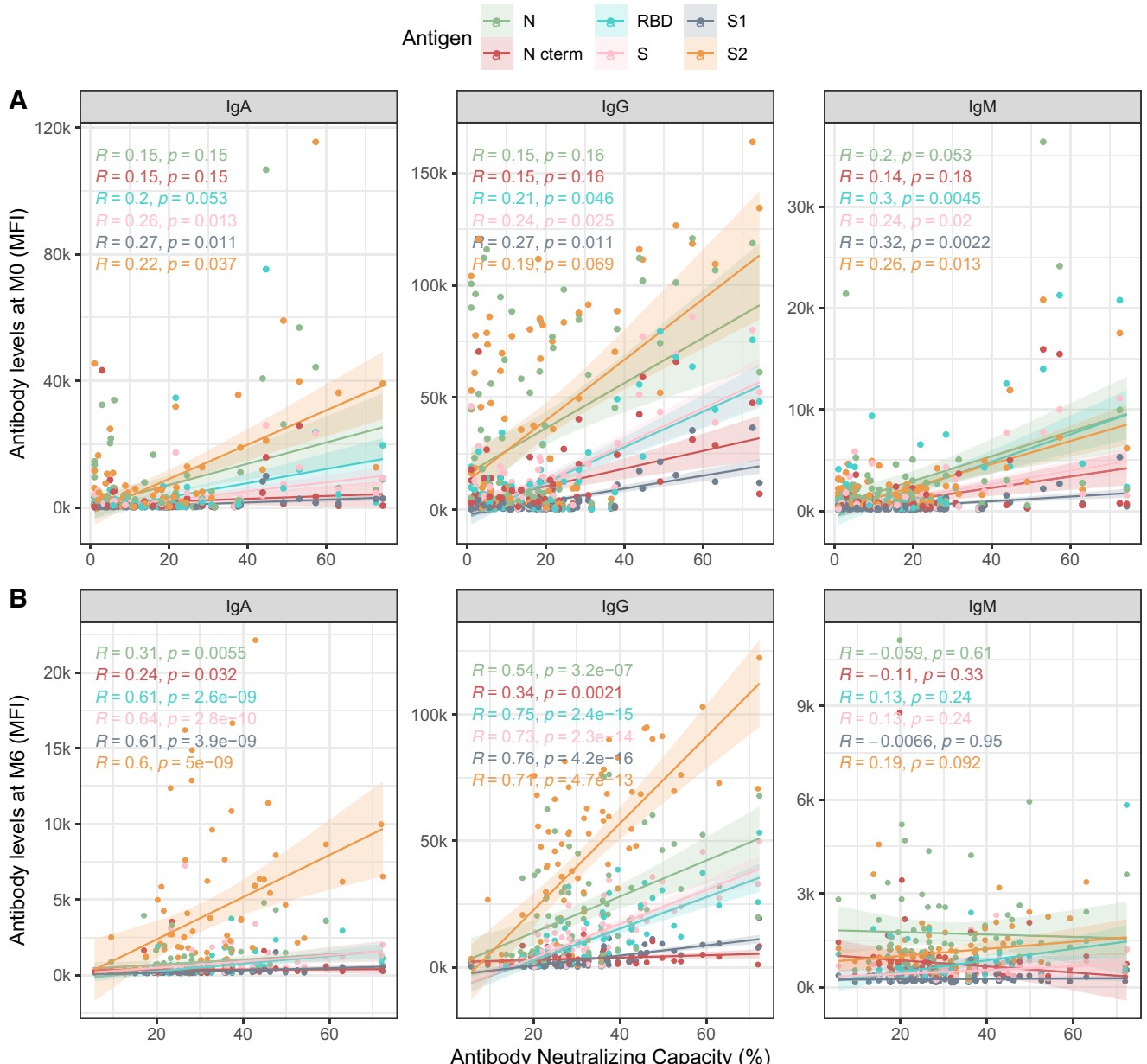

**Fig. 3 Correlations between antibody levels and RBD-ACE2 neutralization capacity.** Spearman's rank correlation test between levels (median fluorescence intensity, MFI) of IgA, IgG, and IgM against each antigen (Nucleocapsid full-length protein (N), and its C-terminal domain, the Receptor Binding Domain (RBD), full S protein and its subregions S1 and S2) at **A** baseline visit (M0) and **B** M6 visit; and plasma neutralization capacity (as a percentage of RBD-ACE2 binding inhibition). Two-sided spearman test was used to calculate the p-values and $r_s$ correlation coefficients are color-coded for each antigen/isotype pair. Colored lines represent the fitted curve calculated using the linear model method. Shaded areas represent 95% confidence intervals.

severe COVID-19 with a back-boosting of antibodies against S2 from betacoronaviruses[32], and N and S from OC43[31]. However, these studies included only hospitalized patients, as opposed to our cohort that included mainly asymptomatic and participants with mild/moderate symptoms. HCoV protective immunity against reinfection has been observed to last around 12 months[29]. Knowing the duration of HCoV protective immunity to reinfection and disease will be key to the understanding of HCoV's role on COVID-19 epidemiology and pathology at population level.

We show a cumulative prevalence of SARS-CoV-2 infection of 19.6% (95% CI 16.4–23.0%) after six months of follow-up

(October 2020). The cumulative prevalence around May 2020, corresponding to our second visit (M1), recalculated here with a wider antigen panel, was 16.8% (CI 95% 13.8–20.1%), similar to other studies in Spanish HCW that ranged between 10.5 and 19.9%[43–45]. Around 28% of the total infections detected throughout the follow-up were newly diagnosed after the first visit (M0), which would reveal that infections in the hospital setting mostly happened within the first pandemic wave. No re-infections were reported in our cohort and this could be related to the induction and maintenance of robust neutralizing antibodies along the study period, in contrast with another study in a cohort

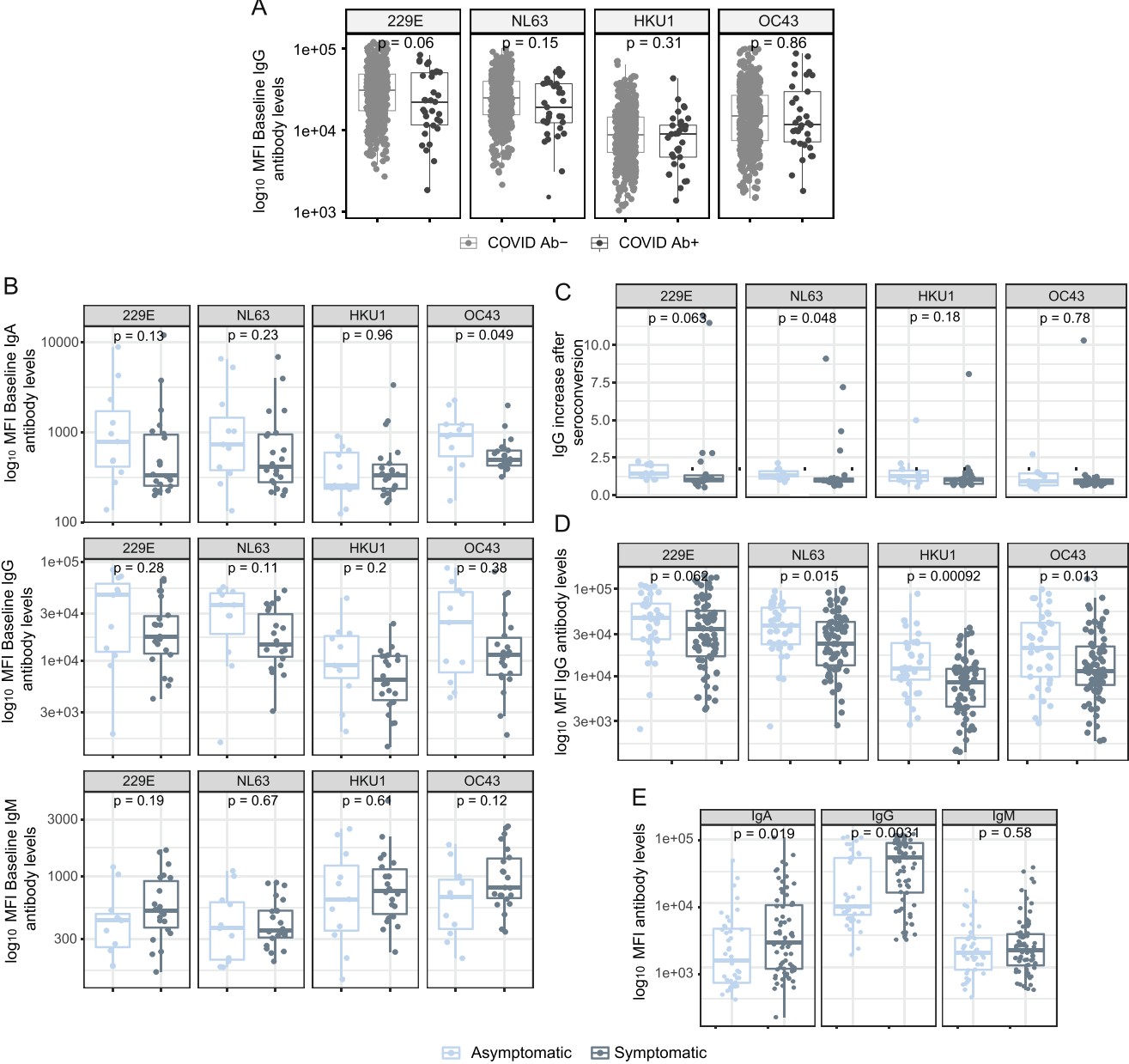

**Fig. 4 The influence of anti-HCoV antibody levels on the antibody response to SARS-CoV-2. A** Differences in baseline levels (median fluorescence intensity, MFI) of IgG against HCoV N proteins between participants who were seronegative during the entire study (COVID-19 Ab-) and participants who seroconverted (COVID-19 Ab+) (n = 468). **B** Differences in IgA, IgG and IgM levels prior to infection against N of the four HCoVs between symptomatic and asymptomatic participants who seroconverted during the study (n = 33). **C** Differences in fold-increase of IgG levels against N of the four HCoVs after SARS-CoV-2 seroconversion in symptomatic vs asymptomatic COVID-19 cases (n = 33). **D** Differences in anti-HCoV N IgG levels at seroconversion visit between symptomatic and asymptomatic SARS-CoV-2 seropositive participants (n = 110). **E** Differences in anti-SARS-CoV-2 N IgG, IgA and IgM levels in asymptomatic versus symptomatic participants at seroconversion visit (n = 110). The center line of boxes depicts the median values; the lower and upper hinges correspond to the first and third quartiles; the distance between the first and third quartiles corresponds to the interquartile range (IQR); whiskers extend from the hinge to the highest or lowest value within 1.5 × IQR of the respective hinge. Two-sided Wilcoxon rank test was used to assess statistically significant differences in antibody levels between groups.

of 173 primary HCW in which 4 reinfections were reported[46]. Surprisingly, only 56% of participants with evidence of infection by serology had a positive rRT-PCR, highlighting that almost half of the infections went under-detected, mainly during the baseline visit (only 49% had a previous positive rRT-PCR) and going up to 73% of rRT-PCR detection rate in the following visits. We observed a high seroreversion rate for IgA and IgM at visits M1 and M3, decreasing at visit M6. This finding reinforces the rapid decay below the seropositivity threshold of these two isotypes

compared to IgG, for which only 9 participants seroreverted between M1 and M6 visits. Although some reports have pointed to a higher antibody decay in HCW with mild symptoms[47], our results show that IgG levels are maintained up to 7.7 months PSO, in line with other studies[2,9,14–16]. Interestingly, IgA levels were maintained in those individuals who did not serorevert during the first 3 months PSO. Furthermore, IgG to N C-terminal rapidly decreased below the positivity threshold, as seen in other studies[2,13–20]. However, the vast majority of participants with a

previous infection remained seropositive for S-related antigens. This finding is of special relevance because RBD and S IgG antibody levels have been shown to correlate with neutralizing activity and S is the main target of currently deployed COVID-19 vaccines and most products under development.

Remarkably, we noticed a pronounced increase in S-related IgG levels from day 150 PSO onwards in 34/46 (73.9%) participants. Previous studies that reached 150 days of follow-up have not highlighted this phenomenon[6,14,16], but it was observed in Figueiredo-Campos et al.[9]. Chen et al.[17] assessed a subset of individuals with stable or increasing antibody levels at day ~100. In our study, nearly all increasers showed the boost in levels for more than one antigen-isotype pair, in line with the results observed by Chen et al.[17]. We also found a consistent tendency pointing to shorter duration of symptoms in participants with higher increase indices, labeled as quick healers, independently from their age. In contrast with their work, we found statistically significant differences in SARS-CoV-2 antibody levels at seroconversion, with decayers showing higher levels compared to sustainers-increasers for N IgG, S2 IgG, and S1 IgM. The increase in antibody levels in recovered participants could be related to a natural boost after a re-exposure, although we do not have any evidence of reinfection, and sustainers/increasers did not report more contacts with positive cases than decayers. A similar late increase in antibody levels has been reported in a study describing immunity to Ebola virus, showing a pattern of decay-stimulation of antibody production in survivors who had been neither re-exposed nor vaccinated, and had been asymptomatic since the infection[48]. The authors argued that the increase in antibodies could be the result of de novo antigenic stimulation at immune-privileged sites, that is, the persistence of antigens in specific organs would mimic a re-infection and boost immunity. Interestingly, Gaebler et al. observed SARS-CoV-2 antigen persistence in the small intestine and related it with the memory B cell response evolving during the first 6 months after infection, with accumulation of Ig somatic mutations, and production of antibodies with increased neutralizing breadth and potency[10].

Strong correlations were found between antibody neutralization capacity and the days PSO, as identified in the previous literature[16,19,20,22], in accordance with the antibody affinity increase after the maturation of the immune response. Anti-spike antigens contributed to an increase in the antibody neutralization capacity, whilst anti-N C-terminal IgG negatively impacted it. IgM may have a neutralization role early after infection but it may be lost after a few months, consistent with the decay of IgM levels. Antibodies from sustainers/increasers and decayers had equivalent neutralization capacities, suggesting that the increasing antibody levels observed 150 days PSO are not associated with the quality of the response. Unexpectedly, IgM increase index negatively correlated with the antibody neutralization capacity at baseline and after six months visits. It would appear that the virus could be more persistent in participants with lower neutralizing capacity and as a result IgM response is successively increased.

The main limitations of this study are that our cohort had few participants with severe disease, and that we only assessed the impact of anti-HCoV N antibodies on SARS-CoV-2 response, while anti-N antibodies are not expected to have neutralizing capacity. However, it is likely that sera with high levels of N HCoV antibodies would also have high levels of antibodies targeting S antigens and B and T cells specific to HCoV, which could explain the potential association with a protective effect. Altogether, further studies will be needed to elucidate the potential role of prior HCoV infections in the spectrum of COVID-19 severity, as well as the temporal relevance of HCoV exposure and the possible impact on vaccine responses.

In conclusion, antibody levels and neutralizing capacity are generally maintained up to 7.7 months, and in a substantial number of individuals antibody levels increase after some months PSO. Further studies are needed to elucidate the mechanisms and nature of these increases and their implications for virus shedding and disease progression. Importantly, previous exposure to HCoVs could have a protective effect against SARS-CoV-2 infection and symptoms development, and may explain in part the differential susceptibility to disease in the population. Additional work focusing on prospective cohorts would allow the assessment of mechanisms and confirm causality in anti-HCoV antibodies on SARS-CoV-2 acquisition, disease progression, immune response maintenance, and correlates of protection.

## Methods

**Study design, population, and setting.** We measured the levels of antibodies to SARS-CoV-2 antigens in blood samples of 578 randomly selected HCW from HCB followed up at four visits: baseline—hereby termed "M0"—(month 0, 28 March–9 April 2020, $n = 578$), "M1" (month 1, 27 April–6 May 2020, $n = 566$), "M3" (month 3, 28 July–6 August 2020) when only participants with previous evidence of infection were invited ($n = 70$), and "M6" (month 6, 29 Sept–20 Oct 2020, $n = 507$) (12.3% lost to follow-up). We collected retrospective data on symptoms through REDCap version 8.8.2 in order to set the beginning of the disease, and the longest period since symptoms onset was 231 days (7.7 months).

The study population included HCW who delivered care and services directly or indirectly to patients, as described[4,42]. We collected nasopharyngeal swabs for SARS-CoV-2 rRT-PCR at M0 and M1 and a blood sample for antibody and immunological assessments at all visits. SARS-CoV-2 detection by rRT-PCR followed the CDC-006-00019 CDC/DDID/NCIRD/ Division of Viral Diseases protocol, as previously described[4,42]. Participants isolated at home due to a COVID-19 diagnosis or on quarantine, were visited at their households for sample and questionnaires collection.

Written informed consent was obtained from all study participants prior to study initiation. The study was approved by the Ethics Committee at HCB (Ref number: HCB/2020/0336). Data for each participant were collected in a standardized electronic questionnaire as described[42].

**Quantification of antibodies to SARS-CoV-2.** IgM, IgG, and IgA antibodies to the full-length SARS-CoV-2 S protein, its subregions S1 and S2, RBD that lies within the S1 region, the N full-length protein, and its specific C-terminal region, and the full-length N protein of the HCoVs HKU1, 229E, OC43 and NL63, were measured by Luminex (Supplementary Information) based on a previously described protocol[49]. Sequential plasma samples from the same individual were tested together in the same assay plate. Assay positivity cutoffs specific for each isotype and analyte were calculated as 10 to the mean plus 3 standard deviations (SD) of $\log_{10}$-transformed mean fluorescence intensity (MFI) of 129 pre-pandemic controls. Results were defined as undetermined when the MFI levels for a given isotype-analyte were between the positivity threshold and an upper limit at 10 to the mean plus 4.5 SD of the $\log_{10}$-transformed MFIs of pre-pandemic samples, and no other isotype-antigen combination was above the positivity cutoff and the participant did not have any previous evidence of seropositivity or rRT-PCR positivity.

**Neutralizing antibodies.** Percentage of inhibition of RBD binding to ACE2 by plasma was analyzed through a flow cytometric-based in vitro assay as detailed in the Supplementary Information. This technique stands for its rapidity and efficiency and sets a potential alternative to the more demanding plaque-reduction neutralization assays. Briefly, a murine stable cell line expressing the ACE2 receptor was incubated with RBD-mFc fusion protein, composed of RBD fused to the Fc region of murine IgG1, previously exposed to the different plasma samples at a dilution 1/50. Cells were stained with anti-mouse IgG-PE, washed, and analyzed by flow cytometry using standard procedures. One hundred and one samples were tested alongside 20 positive and 20 negative pre-pandemic controls, in duplicates (Supplementary Fig. 9a). We cross-validated the neutralization assay with a validated assay[50]. Fifty-five plasma samples were analyzed for pseudovirus neutralization and half-maximal dilutions concentrations (ID50) were compared with the results obtained with the flow cytometry assay. There was a strong correlation (rho = 0.9, $p < 0.0001$) between both assays (Supplementary Fig. 9b).

**Statistical data analysis.** Prevalence of SARS-CoV-2 antibodies or SARS-CoV-2 infection confirmed by rRT-PCR, and cumulative prevalence of past or current infection (positive SARS-CoV-2 rRT-PCR and/or antibody seropositivity at any time point) were calculated as proportions with 95% CI.

We tested the association between variables with the Chi-square or Fisher's exact test for categorical variables, and with the Wilcoxon Sum Rank test for

continuous variables. Paired Samples Wilcoxon Test was used for paired continuous data. We assessed the relationships between continuous variables using linear regression models and Spearman's rank correlation test. Locally estimated scatterplot smoothing (LOESS) was used to visualize trends in antibody levels over days PSO or post rRT-PCR diagnosis.

A Venn diagram was created to illustrate the overlap between anti-N full-length protein, anti-N C-term, anti-RBD, anti-S, anti-S1, anti-S2 in the Sustainer/Increaser groups[51].

Univariable and multivariable linear regression models were run to assess factors associated with SARS-CoV-2 antibody levels and prevalence. The variables tested were the following: sex and age, presence of COVID19 symptoms (individual symptoms also included—fatigue, cough, dyspnea, and other respiratory symptoms, anosmia or ageusia, sore throat, fever, rhinorrhea, headache, chills, and digestive symptoms-), no. of people living in the household and no. of children, worked in a COVID19 ward, type of job (doctor, nurse, administrative), had daily contact with patients, smoking habits, chronic medication, presence of baseline illness, previous contract with a positive COVID19 case.

We additionally explored the association between the SARS-CoV-2 antibody levels and the percentage of neutralization of RBD at month 6 in a principal components analysis (PCA) that included all isotype/antigen pairs. Before the PCA, we confirmed the adequacy of the analysis by testing the colinearity of the variables with the Kaiser-Meyer-Olkin analysis (>0.5) and the Bartlett's sphericity test ($p <$ 0.001). The number of factors chosen was based on eigenvalues >1 that explained >75% of the total variance. To investigate the relationships between HCoV levels and a subset of variables with clinical outcomes and SARS-CoV-2 antibody levels, we built multivariable logistic and linear models, respectively, for those participants for whom we had a sample prior to seroconversion.

A $P$-value of ≤0.05 was considered statistically significant and 95% CIs were calculated for all estimates. We performed the statistical analysis in R version 4.0.3 (packages tidyverse, corrplot, FactomineR, pls, and MASS).

**Reporting summary**. Further information on research design is available in the Nature Research Reporting Summary linked to this article.

## Data availability
The antibody levels, neutralization data generated in this study are deposited in the UB repository under this link: https://doi.org/10.34810/data125. The raw identifying data are protected and are not available due to data privacy laws.

## Code availability
Code used in the analysis is available at https://doi.org/10.34810/data125.

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

## Acknowledgements

We thank the participation of health care workers who are committed to this study as well as with their professional activity that has a positive impact in the society, even more relevant during the pandemic. We are grateful to Eugénia Chóliz, Pau Cisteró, Antía Figueroa-Romero, Silvia Folchs, Jochen Hecht, Mikel J. Martínez, Núria Pey, Patricia Sotomayor and Sara Torres who participated in the field and/or laboratory work during previous visits. We also thank Cristina Castellana and the administrative department in ISGlobal, Sergi Sanz, and Sergio Olmos for statistical advice, Jordi Rello for scientific advice, and the nurses from Occupational Health and Preventive Medicine departments (HCB). This work was supported by Institut de Salut Global de Barcelona (ISGlobal) internal funds; in-kind contributions of Hospital Clínic de Barcelona; European Institute of Innovation and Technology (EIT) Health (Grant number 20877), supported by the European Institute of Innovation and Technology, a body of the European Union receiving support from the H2020 Research and Innovation Programme; Fundació Privada Daniel Bravo Andreu through high-throughput equipment provided. We also acknowledge support from the Spanish Ministry of Science and Innovation through the "Centro de Excelencia Severo Ochoa 2019-2023" Program (CEX2018-000806-S), and support from the Generalitat de Catalunya through the CERCA Program. P.S. work was supported by 2017-SGR-3380 and MINECO RTI2018-093964-B-I00. G.M. was supported by the Departament de Salut, Generalitat de Catalunya (grant number SLT006/17/00109). L.I.'s work was supported by PID2019-110810RB-I00 grant from the Spanish Ministry of Science & Innovation. Work at IrsiCaixa was partially funded by Grifols, the Departament de Salut of the Generalitat de Catalunya (grant SLD016 to J.B.), the Spanish Health Institute Carlos III (Grant PI17/01518 and PI20/00093 to J.B.), CERCA Programme/Generalitat de Catalunya 2017 SGR 252, and the crowdfunding initiatives #joemcorono, BonPreu/Esclat and Correos. E.P. was supported by a doctoral grant from the National Agency for Research and Development of Chile (ANID): 72180406. The funders had no role in study design, data collection, and analysis, the decision to publish, or the preparation of the manuscript.

## Author contributions

A.A., A.L.G.B., P.E., G.M., A.M. and C.D. designed the study. A.C., M.L., N.O., M.R., N.R., R.A., D.B., C.D. and S.W. recruited participants, collected data and obtained samples at the clinic. A.J., M.V., R.A., R.R., D.B., R.A.M., L.P., S.A. and C.J. processed the samples, developed and performed the serological assays and analysis. A.A., P.E. and P.H.L., performed the flow cytometry neutralization assay and E.P., B.T. and J.B. the pseudovirus-based neutralization assay. R.A., R.S., S.B., A.V., A.L., A.T., P.V. and M.T. contributed to design and the critical interpretation of the results. J.C., L.I., N.R.M., C.C., P.Se., D.P. and P.Sa. produced the antigens. N.O. and M.R. analyzed the data and R.S. gave support to data analysis. S.M. managed the clinical data. G.M. and C.D. supervised the antibody assays and data analyses. N.O., M.R., G.M. and C.D. wrote the first draft of the paper. All authors approved the final version as submitted to the journal.

## Competing interests

The authors declare no competing interests.

## Additional information

[1]ISGlobal, Hospital Clínic, Universitat de Barcelona, Barcelona, Catalonia, Spain. [2]Institut d'Investigacions Biomèdiques August Pi i Sunyer, Barcelona, Spain. [3]Immunology Unit, Department of Biomedical Sciences, Faculty of Medicine and Health Sciences, University of Barcelona, Barcelona, Spain. [4]Centro de Investigação em Saúde de Manhiça, Maputo, Mozambique. [5]Spanish Consortium for Research in Epidemiology and Public Health, Madrid, Spain. [6]School of Health Sciences TecnoCampus Universitat Pompeu Fabra, Mataró, Spain. [7]Department of Preventive Medicine and Epidemiology, Hospital Clinic, Universitat de Barcelona, Barcelona, Spain. [8]Biomolecular screening and Protein Technologies Unit, Centre for Genomic Regulation (CRG), The Barcelona Institute of Science and Technology, Barcelona, Spain. [9]IrsiCaixa AIDS Research Institute, Germans Trias i Pujol Research Institute (IGTP), Can Ruti Campus, UAB, Badalona, Catalonia, Spain. [10]University of Vic–Central University of

Catalonia (UVic-UCC), Vic, Catalonia, Spain. [11]Occupational Health Department, Hospital Clínic, Universitat de Barcelona, Barcelona, Spain. [12]Faculty of Medicine and Health Sciences, Universitat de Barcelona, Barcelona, Spain. [13]Julia McFarlane Diabetes Research Centre, Cumming School of Medicine, University of Calgary, Calgary, AB, Canada. [14]Department of Microbiology, Immunology and Infectious Diseases, Snyder Institute for Chronic Diseases, Cumming School of Medicine, University of Calgary, Calgary, AB, Canada. [15]International Health Department, Hospital Clínic, Universitat de Barcelona, Barcelona, Spain. [16]These authors contributed equally: Natalia Ortega, Marta Ribes. [17]These authors jointly supervised this work: Gemma Moncunill, Carlota Dobaño. ✉email: gemma.moncunill@isglobal.org; carlota.dobano@isglobal.org

