## [Peer Review File · Nature Communications]

REVIEWER COMMENTS

Reviewer #1 (Remarks to the Author):

Overall this is a nice study which builds upon previous work by the group in the development of their multiplex assay and uses a good sample set. They investigate the role of endemic CoVs in offering protection to COVID-19 finding protective role for hCoV antibodies. However, I have concerns surrounding their interpretation of the data. It appears to me that their data instead shows a correlation between hCoV infection and progression of COVID-19 (asymptomatic versus symptomatic), which has been previously reported by several other authors, as opposed to any role in protection. Another concern would be assay performance, particularly surrounding controls or correcting for controls. Overall, the discussion should be rewritten to more accurately reflect the data that is presented in the manuscript, namely that there is a correlation in progression, but not in protection against SARS-CoV-2.

1. 2 key references missing: Becker et al (<https://www.nature.com/articles/s41467-021-20973-3>) and Edridge et al (<https://www.nature.com/articles/s41591-020-1083-1>). The first performed used a multiplex assay to investigate hCoV protection offered, finding no correlation, the second shows that immunity to hCoVs are very short (common reinfection inside 12 months).
2. Include demographics as supplementary table instead of citation
3. Sentence "IgG to 229E and NL63, and IgM to 229E and HKU1, significantly increased after SARS-CoV-2 seroconversion." The data presented shows this statement to be false. Only IgG to 229E is significant, everything else is non-significant.
4. Assay controls. What correction was done to account for plate by plate differences in the luminex assay? Although the assay has good performance, has it been compared at any point to a commercial assay (i.e. Roche or Mikrogen)?
5. The new neutralization assay needs to be validated for performance against an some kind of actual viral neutralization test (either a VNT or PRNT). Until it has been validated, you cannot confirm the level of neutralization you have.
6. Also what controls were included for the neutralization assay
7. Can the symptomatic patients be further subdivided into groups based on severity of condition? Would be interesting to see if correlation between asympto and symptomatic also extends to severity in symptomatic donors.
8. Dashed line on Fig 1 needs to be clearer on some panels
9. Fig 2, why is M3 not included, as per the methods blood should have been taken at this time point and it should help fill the gap in the data between 50 and 150
10. For the included hCoVs, what is the sequence similarity in the N's
11. Supp Fig 6, grouping might be clearer if ordered by disease instead of ab subtype

Reviewer #2 (Remarks to the Author):

In "Seven-month kinetics of SARS-CoV-2 antibodies and protective role of pre-existing antibodies to seasonal human coronaviruses on COVID-19", Ortega and colleagues characterize the impact of pre-existing non-SARS-CoV-2-anti-coronavirus antibodies (against the common cold coronaviruses) and their protective effect on patients. This study follows a cohort of health care workers over a 7 month time period and find a steady level of class-switched antibodies and a reduction in symptomatic cases in patients with pre-existing IgG and IgA to common cold coronaviruses, indicating a potentially cross-protective effect.

The authors take on a very timely and complex topic by employing not only antibody titering but also

neutralization (providing a more functional measure of protection), and characterizing all three major antibody subtypes over a longer time course than many other studies, which may help to resolve the occasionally conflicting evidence cited in their introduction. It is also useful that asymptomatic cases were included, which are not often characterized though they are very common. Adding to this, the characterization of pre-existing immunity may help to better understand symptomatic vs asymptomatic cases and allay any fears about potential problems with vaccination interfering with disease response is very significant. On a technical note, the use of flow cytometry to quickly and efficiently measure neutralization should definitely be highlighted as a potential solution to the issue of more labor-intensive plaque-reduction neutralization assays slowing down SARS-CoV-2 research and causing researchers to lean more heavily on basic titer measures.

Major remarks:

- Is there any indication for patients with pre-existing coronavirus antibodies when they might have gotten sick? How long might this cross-protection last?
- A little more detail for the methodology would be helpful up front – cohort size in the abstract and what HCoV antigens are being used for the intro. These are described later of course but would be fitting in the experimental summary as the introduction has just thoughtfully explained the S vs N neutralizing capacities.
- How does the cross-protective/back-boost effect of pre-existing immunity compare quantitatively to the lighter de novo antibody production? Theoretically it should be greater if there is a protective effect from pre-existing immunity, but can these be compared?
- Why is N so helpful for cross protection vs S? Why was S not looked at?
- With regard to sustainers/increasers vs decayers – do they reach different peak antibody/neutralization titers?

Minor remarks:

- Line 45: Please indicate in the abstract the size of the cohort.
- Line 71: “Most recent” should be changed to “more recent”.
- Line 76-66: This sentence makes it sound as though neutralizing antibodies CAUSE higher disease severity. Perhaps change to “lowered disease severity”.
- Line 171: S1 and S2 antigens are not described within results.
- Line 182: “Data now shown” should be “Data not shown” – data does not need to be shown but it might be worth listing the general groups of variables tested in case there is one you hadn’t thought of or might be useful to measure in the future.
- Line 186: RBD is never spelled out.
- Line 194-195: Indicate positive or negative correlation.
- Line 394: “Along” should be “alongside”.

Reviewer #1

Comment	Answer	Actions
Overall this is a nice study which builds upon previous work by the group in the development of their multiplex assay and uses a good sample set. They investigate the role of endemic CoVs in offering protection to COVID-19 finding protective role for hCoV antibodies. However, I have concerns surrounding their interpretation of the data. It appears to me that their data instead shows a correlation between hCoV infection and progression of COVID-19 (asymptomatic versus symptomatic), which has been previously reported by several other authors, as opposed to any role in protection. Another concern would be assay performance, particularly surrounding controls or correcting for controls. Overall, the discussion should be rewritten to more accurately reflect the data that is presented in the manuscript, namely that there is a correlation in progression, but not in protection against SARS-CoV-2.	Thank you for your thorough review and your comprehensive feedback. We report that HCoV levels could be protective against SARS-CoV-2 infection because those who got infected during the study had or tended to have lower pre-infection HCoV levels than those who did not. In addition, we report a protective role against COVID-19 disease because those who were positive for SARS-CoV-2 antibodies and were asymptomatic had higher pre-infection HCoV levels compared to those who were positive and symptomatic. Even after infection, the levels against HCoV were higher in asymptomatics than symptomatics, whereas the contrary was observed for SARS-CoV-2 antibodies, which were higher in symptomatic than asymptomatic individuals. Therefore, we agree that we mostly show a protective effect against the disease rather than infection. We therefore believe that the discussion already reflects this and have not done a major rewriting.	No actions
1. 2 key references missing: Becker et al (https://www.nature.com/articles/s41467-021-20973-3) and Edridge et al (https://www.nature.com/articles/s41591-020-1083-1). The first performed used a multiplex assay to investigate hCoV protection offered, finding no correlation, the second shows that immunity to hCoVs are very short (common reinfection inside 12 months).	Thank you. We have now cited these two publications.	Added references in lines 93 and 96.
2. Include demographics as supplementary table instead of citation	We have done this.	See page 26, Supplementary Table 1.

3. Sentence “IgG to 229E and NL63, and IgM to 229E and HKU1, significantly increased after SARS-CoV-2 seroconversion.” The data presented shows this statement to be false. Only IgG to 229E is significant, everything else is non-significant.	Thank you for this remark.	The sentence has been modified to express that only levels of IgG to 229E significantly increased. See line 231.
4. Assay controls. What correction was done to account for plate by plate differences in the luminex assay? Although the assay has good performance, has it been compared at any point to a commercial assay (i.e. Roche or Mikrogen)?	The Luminex assay was performed in only 6 plates of 384 wells with samples from the same individual run together in the same plate (from M0 to M6). We do include positive, negative and blank controls in each plate for QC/QA purposes and we check variability across plates and potential batch effects. We do not normalize data routinely (unless a major batch effect is found) as in our experience normalization may introduce more noise. In previous studies we have compared levels of antibodies to N antigens in Roche ECLIA vs Luminex and the correlation was very high. We have compared our Luminex assay to Euroimmune and Vircell ELISAs for spike IgG and found a superior performance for the Luminex. We have also data for some samples measured by Mikrogen for a different study but we do not consider it relevant to report here as we did our independent evaluation of the Luminex assays against >100 positive and negative controls following FIND guidelines and reported this separately. To this end, we assessed its performance with samples from individuals with PCR confirmed SARS-CoV-2 infection, with asymptomatic, symptomatic mild or severe infections and with a range of days since infection. The sensitivity and specificity were very high (Dobaño et al., 2021).	See Supplementary Information line 822.
5. The new neutralization assay needs to be validated for performance against an some kind of actual viral neutralization test (either a VNT or PRNT). Until it has been validated, you	We performed additional experiments to compare our flow cytometry neutralization assay to a validated assay based on pseudovirus. The pseudovirus-based neutralization assay uses HIV-based pseudovirus and ACE2 expressing 293T cells and is described in Pradenas et al. Cell Med 202. That assay	See Supplementary figure 9b. Methods section (line

cannot confirm the level of neutralization you have.	had been validated by direct comparison of IC50 neutralization values obtained using pseudoviruses infecting ACE2 expressing 293 cells and replicative viruses infecting Vero cells, in Trinité et al. Sci Repo 2021 Here we used 55 samples that were measured in duplicates with both assays and results had a correlation of 0.9 (p<0.0001). We did this in collaboration with IRSI-Caixa virology lab, and added 3 new coauthors in the paper to account for this extra work.	414) and Supplementary Methods (line 900).
6. Also what controls were included for the neutralization assay	We included 20 positive controls and 20 negative pre-pandemic controls.	We have described in the Methods section (line 413) what controls were included in the neutralization assay. We have also included a dotplot (Supplementary Fig 9a).
7. Can the symptomatic patients be further subdivided into groups based on severity of condition? Would be interesting to see if correlation between asympto and symptomatic also extends to severity in symptomatic donors.	In our cohort, only one participant was hospitalized due to COVID-19. However, we collected their duration up to ten days or more. Categorizing participants according to less and more than 10 days of symptom duration, no significant difference or common trend is observed in HCoV antibody levels prior to SARS-CoV-2 infection evidence (data not shown in the manuscript)	No actions.
8. Dashed line on Fig 1 needs to be clearer on some panels	Thank you for this remark.	Figure has been replaced, although we may need to upload a high

		quality figure, which seems not to be possible during revision of the manuscript.
9. Fig 2, why is M3 not included, as per the methods blood should have been taken at this time point and it should help fill the gap in the data between 50 and 150	Blood samples at M3 were collected by finger prick and plasma volume obtained per participant was not enough to perform de Luminex and the neutralization assays, that is why we do not have neutralization data from this time point.	No actions.
10. For the included hCoVs, what is the sequence similarity in the N's	The amino acid pairwise similarities of full-length SARS-CoV-2 N protein and seasonal HCoVs are 36% to 229E, 39.1% to NL63, 48.1% to OC43 and 47% to HKU1. Amino acid pairwise identities of full-length SARS-CoV-2 N protein and seasonal HCoVs are 26.4% (to 229E), 27% (NL63), 35.7% (OC43) and 35.2% (HKU1) (See figure below). However, these identities are lower in a specific C-terminal immunodominant domain of N, used to discern cross-reactivity with endemic HCoV. This issue is further discussed in Dobaño et al., 2020, where cross-reactivity of antibodies to N protein is explored.	We have added this information in the Results section (page 6 line 225).

	 <caption>Antibody Levels by Virus and Epitope</caption>   Virus EP1 (AAs 50-70) EP2 (AAs 133-207) EP3 (AAs 248-272) EP4 (AAs 348-416) Total     229E (α-CoV) 14.3% 33.9% 36% 9.3% 26.4%   NL63 (α-CoV) 10% 35.7% 41.7% 14.0% 27%   OC43 (β-CoV) 42.9% 47.8% 48% 11.6% 35.7%   HKU1 (β-CoV) 52.4% 41.8% 56% 11.6% 35.2%   	Virus	EP1 (AAs 50-70)	EP2 (AAs 133-207)	EP3 (AAs 248-272)	EP4 (AAs 348-416)	Total	229E (α-CoV)	14.3%	33.9%	36%	9.3%	26.4%	NL63 (α-CoV)	10%	35.7%	41.7%	14.0%	27%	OC43 (β-CoV)	42.9%	47.8%	48%	11.6%	35.7%	HKU1 (β-CoV)	52.4%	41.8%	56%	11.6%	35.2%	
Virus	EP1 (AAs 50-70)	EP2 (AAs 133-207)	EP3 (AAs 248-272)	EP4 (AAs 348-416)	Total																											
229E (α-CoV)	14.3%	33.9%	36%	9.3%	26.4%																											
NL63 (α-CoV)	10%	35.7%	41.7%	14.0%	27%																											
OC43 (β-CoV)	42.9%	47.8%	48%	11.6%	35.7%																											
HKU1 (β-CoV)	52.4%	41.8%	56%	11.6%	35.2%																											
11. Supp Fig 6, grouping might be clearer if ordered by disease instead of ab subtype	We performed a hierarchical clustering of antibody levels to clearly visualize the patterns and correlations between them.	It has been specified in the figure footer (line 787).																														

Reviewer #2

Major Remarks

Comment	Answer	Actions
In “Seven-month kinetics of SARS-CoV-2 antibodies and protective role of pre-existing antibodies to seasonal human coronaviruses on COVID-19”, Ortega and colleagues characterize the impact of pre-existing non-SARS-CoV-2-anti-coronavirus antibodies (against the common cold coronaviruses) and their protective effect on patients. This study follows a cohort of health care workers over a 7 month time period and find a steady level of class-switched antibodies and a reduction in symptomatic cases in patients with pre-existing IgG	We thank you for your revision and thoughtful comments. We have highlighted the use of flow cytometry for neutralization analyses.	See line 407.

and IgA to common cold coronaviruses, indicating a potentially cross-protective effect. The authors take on a very timely and complex topic by employing not only antibody titering but also neutralization (providing a more functional measure of protection), and characterizing all three major antibody subtypes over a longer time course than many other studies, which may help to resolve the occasionally conflicting evidence cited in their introduction. It is also useful that asymptomatic cases were included, which are not often characterized though they are very common. Adding to this, the characterization of pre-existing immunity may help to better understand symptomatic vs asymptomatic cases and allay any fears about potential problems with vaccination interfering with disease response is very significant. On a technical note, the use of flow cytometry to quickly and efficiently measure neutralization should definitely be highlighted as a potential solution to the issue of more labor-intensive plaque-reduction neutralization assays slowing down SARS-CoV-2 research and causing researchers to lean more heavily on basic titer measures		
1. Is there any indication for patients with pre-existing coronavirus antibodies when they might have gotten sick? How long might this cross-protection last?	Unfortunately, we do not have any indication of previous infection episodes nor negative controls for HCoVs to establish a positivity threshold to determine seroreversions and calculate how long they may last. Some studies have reported that HCoVs antibodies protection lasts around 12 months (RW. D. Edridge et al.), but it is a difficult-to-implement design because of the high-prevalence in the general population.	We have included a reference on HCoV immunity protection duration and its implications in the Introduction and the Discussion (line 285).

2. A little more detail for the methodology would be helpful up front – cohort size in the abstract and what HCoV antigens are being used for the intro. These are described later of course but would be fitting in the experimental summary as the introduction has just thoughtfully explained the S vs N neutralizing capacities.	We added the cohort size in the abstract and that we tested the nucleocapsid antigen of four seasonal coronaviruses.	Added (N=578) and nucleocapsid in the abstract and in the introduction (lines 47, 49 and 127).
3. How does the cross-protective/back-boost effect of pre-existing immunity compare quantitatively to the lighter de novo antibody production? Theoretically it should be greater if there is a protective effect from pre-existing immunity, but can these be compared?	Indeed, that's a remarkable point. Unfortunately it is not easy to assess in our study. To test this we would need to define the individuals who have been recently exposed to HCoV but we cannot identify seropositive individuals for HCoV because we do not have the required negative controls. However, in Fig S8, we show how the HCoV antibody levels at baseline are lower when the novo production of SARS-CoV-2 antibody levels is higher. Which would indicate that there is no cross-protective/back-boost effect on SARS-CoV-2 antibodies. Nevertheless, we already know that levels of SARS-CoV-2 antibodies positively correlate with exposure/viral load. Therefore, it makes sense that if immunity to HCoV is protective against COVID-19, the levels of antibodies against SARS-CoV-2 are lower, which is what we observe in our study. When we correlate the relative increase in SARSCoV2 antibody levels to the fold change in HCoV we only see a significant inverse correlation for OC43.	No action.
4. Why is N so helpful for cross protection vs S? Why was S not looked at?	The choice of N was supported by the fact that in our early exploratory analyses we observed high backgrounds for IgG against SARS-CoV-2 N full length but not for S	No action.

	antigens in pre-pandemic samples, which reflected a high cross-reactivity with hCoV. We wrote a paper about this (Dobaño et al., 2021). There is now evidence of cross-reactivity to S2 too, which is also a conserved region across coronaviruses as opposed to S1 and RBD which are highly variant across strains (Aydillo et al.). In our experiments, IgG antibodies to S2 in pre-pandemic samples are the second highest after N, but the specific responses in SARS-CoV-2 individuals are also very high, so signal to noise is better to discriminate seropositivity. In addition, as a pragmatic approach, N protein is easier to be produced (expressed in E. coli) than S (expressed in eukaryotic systems due to the glycosylations).	
5. With regard to sustainers/increasers vs decayers – do they reach different peak antibody/neutralization titers?	Sustainers/increases and decayers did not have differences neither in the neutralization capacity for the 3 isotypes nor in the antibody titers peak.	No action.

Minor remarks

Comment	Answer	Actions
Line 45: Please indicate in the abstract the size of the cohort.	We have included your suggestion.	Included (N=578) in the abstract. See line 49.
Line 71: “Most recent” should be changed to “more recent”.	We have incorporated your suggestion.	See line 76.
Line 76-66: This sentence makes it sound as though neutralizing antibodies CAUSE higher disease severity. Perhaps change to “lowered disease severity”.	We have rephrased the sentence to “Studies up-to-date point that neutralizing antibodies (nAbs) strongly correlate with antibody titers to S16,19-22 and also positively correlate with increased disease severity”.	See line 81.

Line 171: S1 and S2 antigens are not described within results.	We are sorry but we do not understand what is the issue here. S1 and S2 are included in all the figures in the results section.	No action.
Line 182: “Data now shown” should be “Data not shown” – data does not need to be shown but it might be worth listing the general groups of variables tested in case there is one you hadn’t thought of or might be useful to measure in the future.	We incorporated both suggestions.	We have added the following sentence in line 433: “The variables tested were the following: sex and age, presence of COVID19 symptoms (individual symptoms also included - fatigue, cough, disnea and other respiratory symptoms, anosmia or ageusia, sorethroat, fever, rhinorrea, headache, chills and digestive symptoms-) , n° of people living in the household and n° of children, worked in a COVID19 ward, type of job (doctor, nurse, administrative), had daily contact with patients, smoking habits, chronic medication, presence of baseline illness, previous contract with a positive COVID19 case”.
Line 186: RBD is never spelled out.	It is spelled in line 111.	No action.
Line 194-195: Indicate positive or negative correlation.	Positive correlation	We have added “positively”, see line 202.
Line 394: “Along” should be “alongside”.	We incorporated your feedback.	See line 413.

REVIEWER COMMENTS

Reviewer #1 (Remarks to the Author):

To the author - Please rewrite lines 233-249 to more accurately reflect the data shown. In particular, the sentence of lines 242-246 is not shown in the data. Only two isotypes are higher in asymptomatic, IgM is higher in symptomatic and this is data from figure 4b not 4c. Figure 4c shows a general trend toward IgG levels being lower in symptomatic infections post-infection, where only NL63 is significant. Similarly lines 266-268, one hCoV is significant, three are not, so the sentence is currently misleading.

Reviewer #2 (Remarks to the Author):

In the resubmission of Seven-month kinetics of SARS-CoV-2 antibodies and protective role of pre-existing antibodies to seasonal human coronaviruses on COVID-19, authors have adequately addressed all comments from both reviewers.

Reviewer 1 (Remarks to the Author)

Comment	Answer	Action
To the author - Please rewrite lines 233-249 to more accurately reflect the data shown. In particular, the sentence of lines 242-246 is not shown in the data.	Sentence in lines 242-246 refers to data plotted in Supplementary Figure 8.	We have rephrased the paragraph that the reviewer mentioned to: We have referenced Supplementary Figure 8 one sentence before to clearly support our statement (line 249).
Only two isotypes are higher in asymptomatic, IgM is higher in symptomatic and this is data from figure 4b not 4c.	In lines 233-235, we state that only two isotypes (IgA and IgG) are higher in asymptomatic, and that it is presented in Figure 4b.	No action has been taken.
Figure 4c shows a general trend toward IgG levels being lower in symptomatic infections post-infection, where only NL63 is significant. Similarly lines 266-268, one hCoV is significant, three are not, so the sentence is currently misleading.	We analyzed all three isotypes but we had only shown IgG in the figure. We had generalized for the isotype because the same tendency is observed for all hCoV but the reviewer is right that only IgG against NL63 reached statistical significance level.	We now exclusively comment on the results shown in Figure 4c that include IgG and have made clearer that although there is a common trend, only IgG against NL63 reaches significance level: “Consistently, levels of all three isotypes against alpha HCoVs and of IgA to OC43 experienced a higher fold-increase after SARS-CoV-2 infection in asymptomatic than symptomatic seroconverters (p<0.05)” by “Consistently, levels of IgG against NL63 experienced a higher fold-increase after SARS-CoV-2 infection in asymptomatic than symptomatic seroconverters” Finally, in line 266, we have removed the term “significantly” for coherence with our

		observations.
--	--	---------------

Reviewer 2 (Remarks to the Author)

Comment	Answer	Action
In the resubmission of Seven-month kinetics of SARS-CoV-2 antibodies and protective role of pre-existing antibodies to seasonal human coronaviruses on COVID-19, authors have adequately addressed all comments from both reviewers.	Thank you for the review.	-